# Genome-Wide Association Study Revealed SNP Alleles Associated with Seed Size Traits in African Yam Bean (*Sphenostylis stenocarpa* (Hochst ex. A. Rich.) Harms)

**DOI:** 10.3390/genes13122350

**Published:** 2022-12-13

**Authors:** Oluwaseyi E. Olomitutu, Rajneesh Paliwal, Ayodeji Abe, Olubusayo O. Oluwole, Olaniyi A. Oyatomi, Michael T. Abberton

**Affiliations:** 1Department of Crop and Horticultural Sciences, University of Ibadan, Oduduwa Road, Ibadan 200132, Nigeria; 2Genetic Resources Center, International Institute of Tropical Agriculture, Oyo Road, Ibadan 200001, Nigeria

**Keywords:** African yam bean, phenotypic variation, genomic-assisted breeding, marker-trait association, seed size traits, SNP alleles

## Abstract

Seed size is an important yield and quality-determining trait in higher plants and is also crucial to their evolutionary fitness. In African yam bean (AYB), seed size varies widely among different accessions. However, the genetic basis of such variation has not been adequately documented. A genome-wide marker-trait association study was conducted to identify genomic regions associated with four seed size traits (seed length, seed width, seed thickness, and 100-seed weight) in a panel of 195 AYB accessions. A total of 5416 SNP markers were generated from the diversity array technology sequence (DArTseq) genotype-by-sequencing (GBS)- approach, in which 2491 SNPs were retained after SNP quality control and used for marker-trait association analysis. Significant phenotypic variation was observed for the traits. Broad-sense heritability ranged from 50.0% (seed width) to 66.4% (seed length). The relationships among the traits were positive and significant. Genome-wide association study (GWAS) using the general linear model (GLM) and the mixed linear model (MLM) approaches identified 12 SNP markers significantly associated with seed size traits across the six test environments. The 12 makers explained 6.5–10.8% of the phenotypic variation. Two markers (29420334|F|0-52:C>G-52:C>G and 29420736|F|0-57:G>T-57:G>T) with pleiotropic effects associated with seed width and seed thickness were found. A candidate gene search identified five significant markers (100026424|F|0-37:C>T-37:C>T, 100041049|F|0-42:G>C-42:G>C, 100034480|F|0-31:C>A-31:C>A, 29420365|F|0-55:C>G-55:C>G, and 29420736|F|0-57:G>T-57:G>T) located close to 43 putative genes whose encoding protein products are known to regulate seed size traits. This study revealed significant makers not previously reported for seed size in AYB and could provide useful information for genomic-assisted breeding in AYB.

## 1. Introduction

Meeting global food and nutritional demands under limited growing space and changing climatic conditions is now the major target in crop breeding programs. Recently, the focus has shifted to relatively unknown indigenous legumes such as African yam bean (AYB) due to its dietary protein and mineral content and adaptive nature to wide climatic and soil conditions [1,2]. African yam bean is economically the most important species in the genus Sphenostylis and the most important tuberous legume of tropical Africa [3,4]. The utilization of its food substances (tubers and pulse) is a feature of cultural diversity in Africa [4,5]. The tuber contains, on average, 15.5% crude protein, 1.3% crude fat, and 68.3% carbohydrate [6]. The seed contains 22.5% protein, 53.7% carbohydrate, and 3.6% crude fat content [7]. Despite its numerous other benefits, low seed yield is one of the major constraints to its production [8,9] due to the absence of improved varieties and a lack of significant research attention.

Yield is a complex trait, difficult to improve directly. It is an expression of several component traits that are highly subject to environmental influences [10]. In crop breeding, the greatest yield improvements are associated with the selection and optimization of its component traits. Since the beginning of domestication in agriculture, increased seed size has been a major target as an important yield component trait [11]. Seed size is crucial to plant fitness in crops whose main mode of propagation is by seed and a key factor affecting eating quality and tolerance to abiotic stresses [12,13,14,15]. Compared with small seeds, large seeds accumulate sufficient nourishing substances for faster germination and stronger seedlings that can better compete for light and nutrition and have stronger tolerance to abiotic stresses [12,15,16]. Small seeds, on the other hand, are efficient at dispersal and colonization [17,18]. Seed size is the most commercially valued trait in dry grain legumes, and it varies widely among AYB accessions [4,19,20]. However, the genetic basis for variation in seed size in AYB is not yet known.

Recent advances in high-throughput genomic platforms have created the opportunity for the genome-wide-level understanding of the genetic basis of variation in complex traits at a finer resolution. Association mapping (AM), originally developed for use in mapping human disease genes [21], is now a popular method of maker-trait association studies in plants. Association mapping detects linkage disequilibrium between genetic markers and genes controlling the trait of interest by exploiting recombination events accumulating over many generations in natural populations [22]. It evaluates whether certain alleles within a population are found with specific phenotypes more frequently than expected [23]. Once genes and/or loci are identified and validated, they could be fixed to develop improved genotypes. Association mapping has several advantages over traditional linkage mapping. These include an increased resolution, a reduced research time (using existing populations rather than generating population via biparental crosses), and a higher allele number detection per locus as opposed to only two [24,25,26]. Association mapping also suffers some shortcomings, such as the detection of false positives in the population structure, which is a result of the linkage between causal and noncausal sites, more than one causal site, and epistasis. However, advancement in statistical methods has helped to reduce the rate of false positives [27]. Several GWAS reports have identified putative QTLs/genes in many leguminous crops [28,29,30,31,32,33] that played an important role in understanding the inheritance of quantitative traits [34,35] and trait deployment using a marker-assisted selection approach [36,37]. In legumes, several QTLs have been identified for seed size traits in cowpea [11], soybean [38], and common bean [39,40]. Unfortunately, very limited molecular research has been conducted in indigenous African legumes, including AYB. In AYB, the only available report of association mapping is the preliminary assessment for nutritional qualities by Oluwole et al. [41]. Here, a maker-traits association study was conducted to investigate the genetic basis of variation in seed size traits (seed length, seed width, seed thickness, and 100-seed weight) in AYB using diversity array technology sequence (DArTseq) genotype-by-sequencing (GBS)-based single nucleotide polymorphism (SNP) markers.

## 2. Materials and Methods

### 2.1. Germplasm

The germplasm consisted of 196 AYB accessions obtained from the existing collection of landraces at the Genetic Resource Center, International Institute of Tropical Agriculture, Ibadan, Nigeria. The passport data of the accessions can be found in Olomitutu et al. [9].

### 2.2. Phenotyping

The accessions were phenotyped under optimal field conditions across three IITA research farms in Nigeria—Ibadan, Kano, and Ubiaja—during the 2018 and 2019 cropping seasons. The experimental design was a 14 × 14 α lattice with three replications. Each experimental unit consisted of 4 m single-row plots, with an inter-row spacing of 0.75 m and intrarow spacing of 0.5 m. Phosphorus fertilizer application in the form of triple superphosphate at a rate of 50 kg P/ha and staking were performed three weeks after planting. Manual weeding was carried out when necessary to keep the field clean. Details of the phenotyping methodology are described in Olomitutu et al. [9]. At harvest, data were recorded on 100-seed weight (g), seed length (mm), seed width (mm), and seed thickness (mm) on a plot basis using IITA descriptors for AYB [42].

### 2.3. Genotyping and Quality Control

Leaf samples were collected from three-week-old seedlings of each 196 AYB accession and stored at −80 °C. Genomic DNA (gDNA) was extracted using the diversity array technology (DArT) DNA extraction protocol (https://ordering.seqart.net/files/DArT_DNA_isolation.pdf accessed on 20 January 2019). The isolated gDNA were qualified on 1% agarose gel electrophoresis and quantified using a Nanodrop 2000 spectrophotometer (Thermo Scientific, Waltham, MA, USA) following the manufacturer’s protocol. The high-quality DNA (100 ng/µL) samples were shipped to DArT Pty Ltd., Canberra, Australia, for genotyping using the whole genome profiling service of DArTseq technology [43]. Detailed methodology on complexity reduction, cloning, library construction, and cleaning was described by Egea et al. [44].

A raw dataset of 5416 DArTseq SNPs was generated. The DArTseq SNPs were filtered using call rate ≥ 70%, average reproducibility ≥ 95%, minor allele frequency (MAF) ≥ 0.01, and 20% missing SNP data to remove poor-quality SNPs. After SNP quality control, a total of 2491 SNPs from 195 accessions were retained and used for the genome-wide association study (GWAS). Accession TSs-442 was filtered out due to low-quality SNPs. In the absence of the AYB genome, trimmed sequences of filtered SNPs were aligned on the common bean reference genome v1.1 (available at https://phytozome-next.jgi.doe.gov, (accessed on 20 January 2019)).

### 2.4. Phenotypic Data Analysis

Based on plot means across test locations, a combined analysis of variance (ANOVA) was performed using PROC GLM in SAS [45], with the RANDOM statement and TEST option. Location and year were considered fixed, while all other factors were regarded as random effects. Broad-sense heritability (H^2^) was estimated from the phenotypic (σ^2^_p_) and genotypic (σ^2^_g_) variances [46] and categorized as low (0–30%), moderate (30–60%), and high (>60%) according to [47]. Correlation analysis among the traits was performed using the corPlot function in R [48]. Distribution plots were also constructed for traits using the hist function in R. The plot means of the remaining 195 accessions (in each and combined locations) were used to calculate the best linear unbiased estimate (BLUE) using META-R version 6.04 [49]. The calculated BLUE value of each genotype was further used in the GWAS analysis.

### 2.5. Genome-Wide Association Study and Candidate Gene Identification

The 2491 filtered SNPs from 195 accessions were used for the assessment of population structure and GWAS using TASSEL 5 [50]. The population structure and relatedness analysis among genotypes in the AYB population were conducted using the genomic principal component analysis (PCA) matrix (P) and kinship matrix (K) [51,52]. Marker-trait associations were determined using two different models based on the estimated BLUEs for phenotypic traits (in each and combined locations) and filtered SNPs: the general linear model (GLM) with PCA as the fixed effect (GLM + PCA) and the mixed linear model (MLM) mixed linear model (MLM + PCA + K) [39]. Based on the distribution of *p*-values for traits, marker-trait associations were declared significant at *p*-values of ≥10^−4^ [53,54]. Manhattan and quantile–quantile (Q–Q) plots were constructed using CMplot package in R.

The SNP markers that were significantly associated with seed size traits through GWAS in the combined location analysis were annotated for candidate gene identification in the LIS-legume information systems (https://www.legumeinfo.org accessed on 4 April 2020). Since AYB currently lacks a reference genome, which is a limitation to candidate gene mapping, the genome of a related legume, the common bean (*Phaseolus vulgaris*), was used. A blast search was performed for trimmed nucleotide sequence (60–80 bps) of significant AYB SNPs on the *Phaseolus vulgaris* genome database (*Phaseolus vulgaris* G19833 genome v2.0) in the legume information system. Synteny of related legumes (*Glycine max* 2.0, *Vigna angularis* 3.0, and *Cajanus cajan* 1.0) was also included in the search. After marking the annotated position in the *Phaseolus vulgaris* genome database, the scroll was zoomed to 1 Mb (500 Kbp up and downstream from the annotated position of the AYB SNP tag in the *Phaseolus vulgaris* genome database) to check for the surrounding candidate genes and their encoding protein products and know if they regulate the traits of interest. Identified putative candidate genes in the *Phaseolus vulgaris* genome database were also subsequently further researched in the previous crop studies literature for verification.

## 3. Results

### 3.1. Phenotypic Evaluations

The ANOVA across the test locations revealed significant accession, location, location × year, accession × location, and accession × location × year effects for the four seed size traits. The accession × year interactions effect was significant only for 100-seed weight and seed length, while year effect was significant for seed length and seed width. High broad-sense heritability (66.4%) was obtained in seed length, while seed thickness (57.8%), 100-seed weight (51.6%), and seed width (50.0%) had moderate heritability estimates (Table 1).

The distribution of the seed size traits evaluated in the six environments and the correlation coefficients between the traits are presented in Figure 1 and Figure 2. The relationships among the traits were positive and significant, and the distribution of traits was near-normal.

### 3.2. Genotyping and SNP Filtering

A diverse set of 196 AYB accessions were genotyped using a high-depth DArTseq SNP approach and generated a total of 5416 SNPs. After SNP quality control (call rate ≥ 0.70, marker reproducibility ≥ 0.95, MAF ≥ 0.01, and missing rate ≤ 0.20) (Appendix A), a total of 2491 SNPs were retained and used for GWAS analysis. The MAF ranged between 0.01 and 0.49, with an average of 0.16 (Appendix A). The average heterozygosity of the population was 0.15, while TSs-137 accession (0.006) and TSs-10 (0.41) showed minimum and maximum heterozygosity in the AYB population (Appendix A). The average proportion of missing data (based on genotypes) was 0.012 in the AYB population (Appendix A). Out of 2491 AYB SNPs, only 422 showed genome-wide syntenic relationships with the common bean reference genome. In the common bean genome, most SNPs were aligned on chromosome 2 (58 SNPs), followed by chromosome 3 (51 SNPs), and the least on chromosome 10 (13 SNPs) (Appendix A).

### 3.3. Association Analysis

Principal component analysis revealed that the first three principal components (PCs) respectively accounted for 5.9, 4.8, and 3.7% of the variation among the AYB accessions. No clear clustering (population structure) could be deduced among the accessions based on these two PCs (Figure 3). The coefficient of relatedness in the pairwise kinship matrix ranged between −0.33 and 2.52. The kinship heatmap plot was developed to visualize the relatedness within the population, which indicated low relatedness between accessions (Figure 3).

A total of 58 marker-trait associations were detected for the four seed size traits at a threshold of *p*-values of ≥10^−4^ in each and combined locations analyses (Table 2, Appendix A). The combined test environments analysis revealed that 12 significant SNP markers were associated with all four seed size traits, seven of which were codetected by the two statistical methods (GLM and MLM) used. The 12 SNPs explained 6.5–10.8% of the phenotypic variation. In the GLM, six SNPs were associated with 100-seed weight, four with seed thickness, three with seed width, and one with seed length. One SNP marker each was associated with 100-seed weight and seed length, three with seed width, and four with seed thickness in the MLM. Two markers (29420334|F|0-52:C>G-52:C>G and 29420736|F|0-57:G>T-57:G>T) with pleiotropic effects were both associated with seed width and seed thickness. Six out of the twelve markers were significant in at least one of the location and combined location analyses for the same traits (Table 2). Manhattan and Q–Q plots of the SNP-based associations mapping for the four traits based on GLM and MLM are presented in Figure 4 and Figure 5, respectively. The observed *p*-values for all traits aligned with expected *p*-values, as shown by the Q-Q plots.

Evaluations in Ibadan over the 2 years revealed significant associations among 30 SNP markers with the four seed size traits (Appendix A). Seventeen of these SNPs were codetected by the two statistical methods. The markers explained 5.9 to 10.6% of the observed variance. Two markers with pleiotropic effects, 29421951|F|0-37:A>C-37:A>C (associated with 100-seed weight and seed width) and 100006540|F|0-20:T>G-20:T>G (associated with seed thickness and seed width), were found. In Kano, 14 markers displayed significant associations with all four seed size traits, 11 of which were codetected by both statistical methods. The variance explained by these markers ranged from 7.1 to 9.7%. Three markers, 29422320|F|0-37:T>C-37:T>C (associated with 100-seed weight and seed length), 29421428|F|0-9:C>A-9:C>A (associated with seed length and seed width) and 29420736|F|0-57:G>T-57:G>T (associated with seed thickness and seed width), were found to have pleiotropic effects. In Ubiaja, 12 markers, half of which were codetected by both statistical methods, were significantly associated with all traits. The contribution of all the markers to the phenotypic variation ranged from 5.0 and 10.5%. Evaluations in Ubiaja did not reveal any marker with a pleiotropic effect. Two marker overlaps were found: 29421549|F|0-25:A>C-25:A>C (for HSW in Kano and Ubiaja) and 29420736|F|0-57:G>T-57:G>T (for ST in Ibadan and Kano). Chromosome positions were not given because AYB is yet to have a reference genome (Appendix A).

Candidate gene analysis was performed by blasting trimmed nucleotide sequences (60–80 bps) of significant AYB SNPs in the combined location analysis on the *Phaseolus vulgaris* genome database (*Phaseolus vulgaris* G19833 genome v2.0) in the legume information system. Forty-three candidate genes were identified. These genes were located near (less than 500 kbp) five SNP markers (100026424|F|0-37:C>T-37:C>T, 100041049|F|0-42:G>C-42:G>C, 100034480|F|0-31:C>A-31:C>A, 29420365|F|0-55:C>G-55:C>G, and 29420736|F|0-57:G>T-57:G>T) associated with the four AYB seed traits (Figure 6, Table 3). The five SNP markers were located on chromosomes 1, 2, 6, and 7 of *Phaseolus vulgaris*. The 43 candidate genes have 34 encoding protein products, with some having similar encoding protein products (Table 3). The encoding gene products are known to regulate seed development (UDP-glycosyltransferase superfamily protein, RING-H2 finger protein 2B), seed/fruit size (cytochrome P450 superfamily protein, pentatricopeptide repeat (PPR) superfamily protein; ovate family protein 13), seed weight (cyclin-dependent kinase inhibitor family protein; ATP-binding ABC transporter), seed length (β-carotene isomerase D27), and grain shape (serine/threonine protein phosphatase 2A) in field crops.

## 4. Discussion

The significant differences observed among the accessions for the seed size traits revealed the existence of adequate genetic variability among them. Seeds of AYB are known to harbor vast genetic variability in color, shape, and size [4,19,20,87]. Adewale et al. [4] had earlier suggested the use of six seed characters (seed length, width, and thickness and their ratios) as unique indices for discriminating among AYB accessions. The significant accession × location × year effects for all traits indicated the distinctiveness of the environments in discriminating among the accessions. Moderate to high heritability estimates observed for all traits implied increased power of SNP detection in the accession, hence, identification of true associations between a marker and putative gene [53]. The significant positive correlations between pairs of traits suggest the possibility of simultaneous improvements in the traits to enhance seed yield.

At the genetic level, differences in the extent of relatedness among individuals in a population used for association mapping can lead to the formation of a population structure that can cause spurious associations between genotypes and the traits of interest [88]. Principal component analysis is a widely used multivariate statistical approach proposed by Price et al. [89] that can calculate population relatedness and count groups in a population in order to reduce dimensional genotype data and control population structure (by selecting the first few PCs that present most of the total variation among individuals based on their SNP data). In this study, however, population structure analysis using the PCA approach showed barely noticeable differentiation among accessions. This result of subtle population structure was also confirmed by the multidimensional scaling (MDS) approach using Tassel software. The first three principal components of PCA had a high correlation with the first three principal components of MDS (data not shown). This might have resulted from the fact that most of the accessions, especially the large proportion whose origins are unknown (102 accessions), are possibly from Nigeria. Association analyses between specific phenotypes and genotypes within a genome are an important step toward the discovery of genes controlling the traits [53,90]. Of the 195 accessions used in this study, 137 have been previously utilized for conducting genome-wide association studies for nutritional traits, and several significant SNPs were found to be associated with the studied traits [41]. In this study, model fitness for the GWAS was confirmed by the Q–Q plots. The alignment of observed and expected *p*-values in the Q–Q plots for all the measured traits indicated that spurious associations as a result of population structure and familial relatedness were largely corrected. Using two alternative GWAS models, we found that the MLM model had a stricter decrease in the number of significant markers than the GLM model. This is because GLM is considered a naive model with a high rate of false positives because it does not take population structure into account [91], whereas MLM takes population structure into account and avoids spurious associations [92,93]. Though subtle population structure was found in this study, both models were reported because this was the first attempt at dissecting the genetic basis of seed size traits in AYB using GWAS. Significant QTLs associated with agronomic traits have also been reported in the absence of population structure using GLM and MLM models of GWAS in rice and faba bean crops [92,94]. The contribution of all the significant markers to the phenotypic variation, which ranged from 5.0 to 10.4%, suggested that the markers could be useful for marker-assisted selection in AYB improvement. However, due to the nonexistence of a reference genome for AYB, the exact locations of the markers on the chromosomes remain unknown. The Alliance for Accelerated Crop Improvement in Africa (ACACIA) is currently undertaking the whole genome sequencing of AYB [95]. Markers with pleiotropic effects could be useful in the simultaneous improvement of the correlated traits. The six significant markers that were consistent in the one location and the combined location analyses for the same traits in this study could be considered putative makers.

In this study, the common bean genome was used to align AYB SNPs, and 17% of the filtered 2491 SNPs aligned widely on common bean chr01 to chr11, including six SNPs on scaffolds. These suggest that the two crops have a syntenic relationship, which might be due to their close evolutionary relationship. Similar results have been reported for another African indigenous legume (Bambara groundnut) having syntenic relationships with other legumes (common bean, adzuki bean, and mung bean) [96,97]. The five significant markers found in *Phaseolus vulgaris* genome at a location close to genes whose encoding proteins had been reported in other crops to regulate the same traits as those with which they are associated in AYB can also be considered candidate makers. For example, markers (29420365|F|0-55:C>G-55:C>G and 29420736|F|0-57:G>T-57:G>T) associated with seed length, seed width, and seed thickness were found located close to genes (Phvul.001G153000 and Phvul.007G269900) having ovate family protein 13 as their encoding protein. Ovate family protein 13 is known to regulate fruit shape and seed size in tomato and rice, respectively [80]. Likewise, the pentatricopeptide repeat (PPR) superfamily protein, the encoding protein of genes (Phvul.006G047700, Phvul.006G047700, Phvul.002G024100, Phvul.001G153700, and Phvul.007G278500) located close to markers (100026424|F|0-37:C>T-37:C>T, 100041049|F|0-42:G>C-42:G>C, 100034480|F|0-31:C>A-31:C>A, 29420365|F|0-55:C>G-55:C>G, and 29420736|F|0-57:G>T-57:G>T) associated with the four seed size and shape traits of AYB is known to regulate seed physical traits by participating in RNA intron splicing during seed development [66]. Though significant marker-trait associations were detected in this study, the result serves as a foundation for the genetic understanding of putative makers underlying seed size traits in AYB. The identified significant makers could be targeted by plant breeders in marker-assisted selection to accelerate the genetic improvement of AYB.

## 5. Conclusions

To the best of our knowledge, this study is the first attempt at dissecting the genetic basis of seed size traits in AYB using genome-wide association mapping. Several SNPs were significantly associated with seed size traits in AYB. The five significant SNP markers (100026424|F|0-37:C>T-37:C>T, 100041049|F|0-42:G>C-42:G>C, 100034480|F|0-31:C>A-31:C>A, 29420365|F|0-55:C>G-55:C>G, and 29420736|F|0-57:G>T-57:G>T) found on the *Phaseolus vulgaris* genome should be regarded as candidate markers. It is, therefore, recommended that efforts should be directed toward the validation of the identified significant makers using several mapping populations before they can be targeted for use in marker-assisted selection for seed size traits in AYB.

## Figures and Tables

**Figure 1 genes-13-02350-f001:**
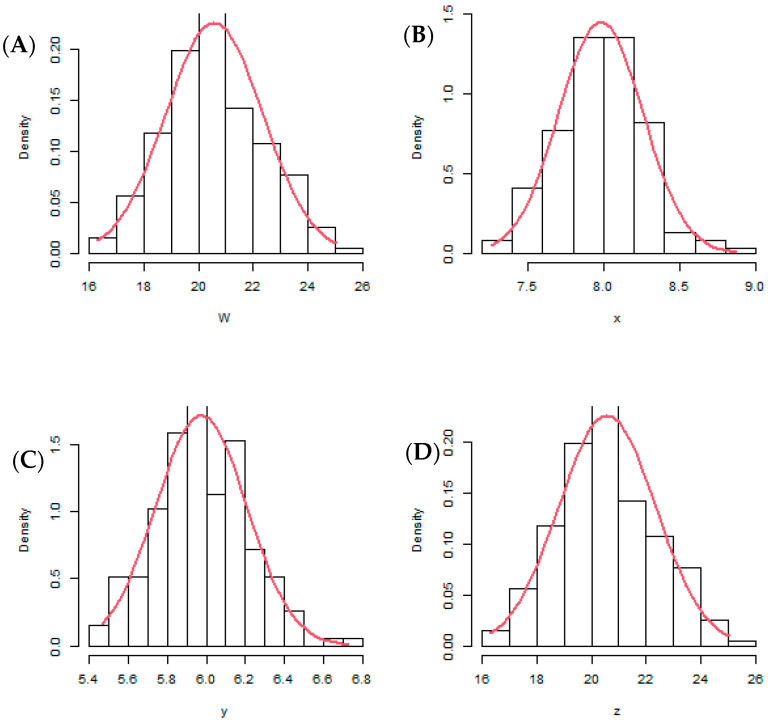
Distribution of seed size traits of 196 accessions of African yam bean evaluated during the 2018 and 2019 cropping seasons at three locations in Nigeria: (**A**) hundred seeds weight (g), (**B**) seed length (mm), (**C**) seed thickness (mm), and (**D**) seed width (mm).

**Figure 2 genes-13-02350-f002:**
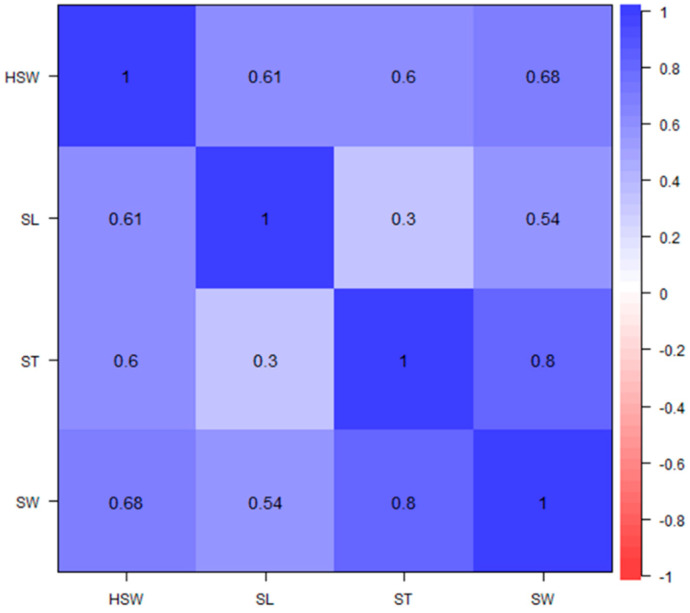
Phenotypic correlation coefficient of seed size traits of 196 accessions of African yam bean evaluated during the 2018 and 2019 cropping seasons at three locations in Nigeria: HSW, 100-seed weight; SL, seed length; SW, seed width; ST, seed thickness.

**Figure 3 genes-13-02350-f003:**
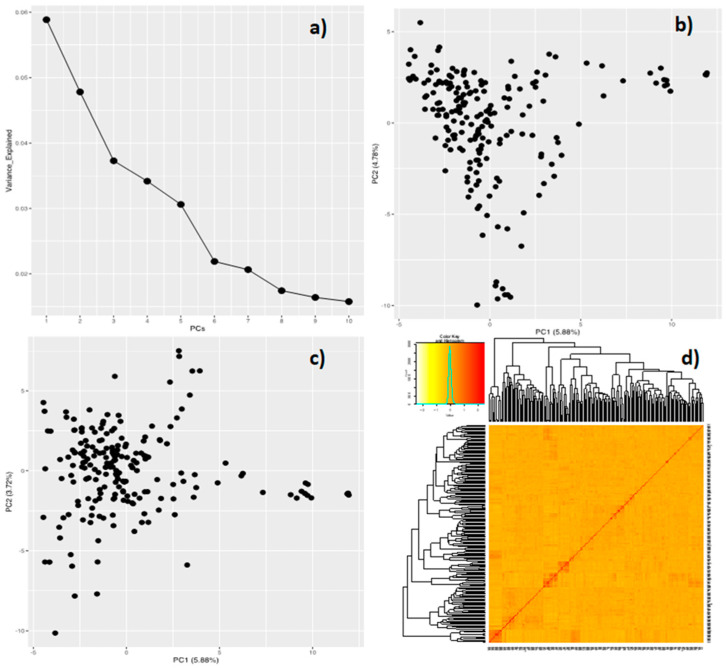
Principal component analysis (**a**–**c**) of AYB population and heatmap plot of kinship matrix (**d**) of 195 African yam bean accessions genotyped with the 2491 DArTseq SNPs.

**Figure 4 genes-13-02350-f004:**
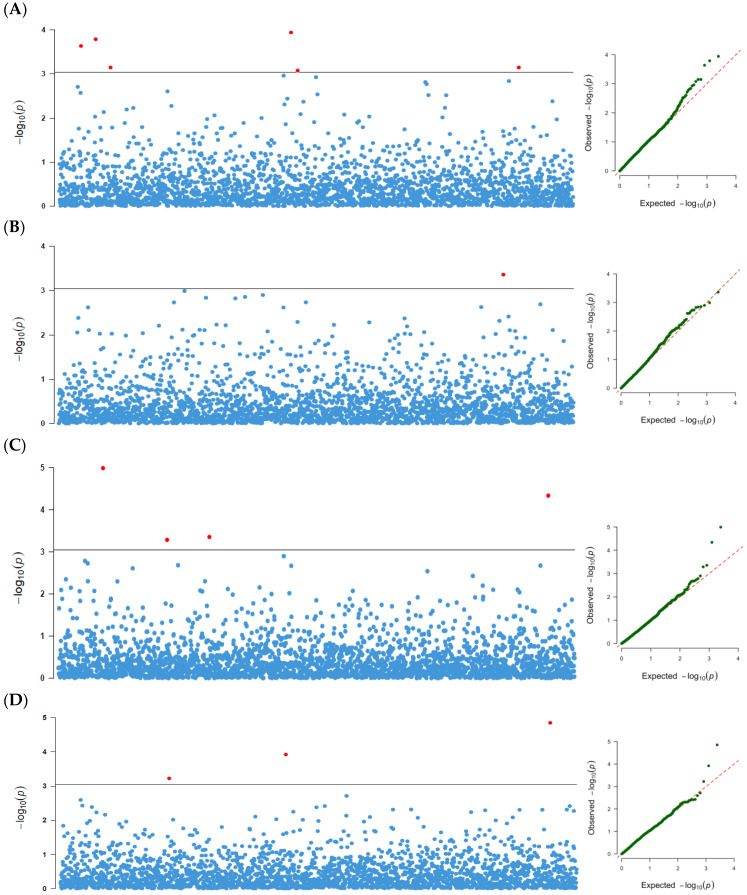
The Manhattan plots and Q–Q plots of the DArTSeq SNP-based associations mapping in the combined test environments analysis for: (**A**) hundred seeds weight (HSW), (**B**) seed length (SL), (**C**) seed thickness (ST), and (**D**) seed width (SW) using the GLM approach.

**Figure 5 genes-13-02350-f005:**
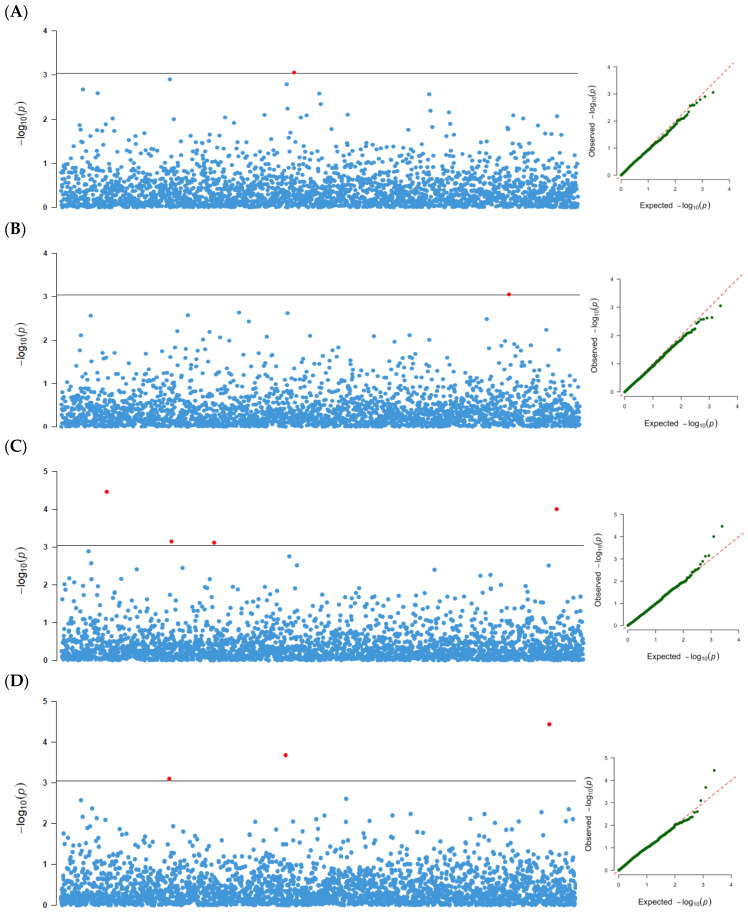
The Manhattan plots and Q–Q plots of the DArTSeq SNP-based associations mapping in the combined test environments analysis for: (**A**) hundred seeds weight (HSW), (**B**) seed length (SL), (**C**) seed thickness (ST), and (**D**) seed width (SW) using the MLM approach.

**Figure 6 genes-13-02350-f006:**
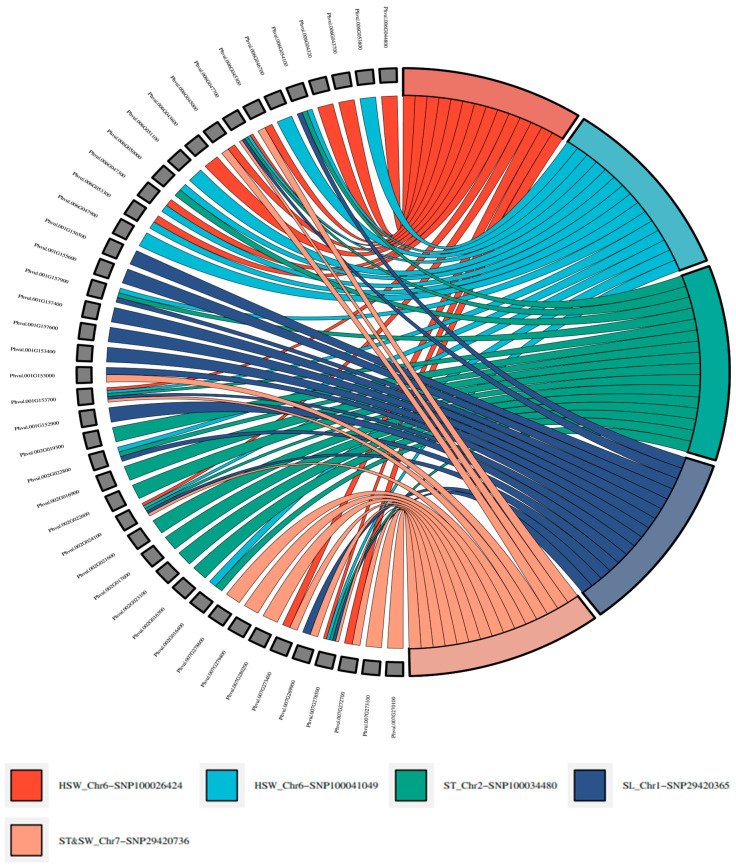
Cord diagram 43 of candidate genes (based on their encoded protein products) associated with five SNP markers associated with seed size traits in the AYB population.

**Table 1 genes-13-02350-t001:** Mean squares from analysis of variance for four seed size traits of 196 accessions of African yam bean evaluated at three test locations in Nigeria.

SOV	DF	HSW	SL	SW	ST
Accession	195	52.18 **	1.26 ***	0.55 **	0.88 ***
Location	2	5863.2 ***	117.51 ***	126.96 ***	148.95 ***
Year	1	78.6	23.45 ***	22.8 ***	0.36
Location × Year	2	536.66 **	19.46 ***	4.32 **	9.39 ***
Accession × Location	390	29.7 **	0.49 ***	0.32 **	0.45 **
Accession × Year	195	28.1 **	0.46 **	0.25	0.31
Accession × Location × Year	390	20.74 ***	0.33 ***	0.23 ***	0.33 ***
Replication (Location × Year)	12	66.67 ***	0.54 **	0.37 **	0.52 **
Block (Replication × Location × Year)	234	17.14 *	0.22	0.14	0.2
Error	2107	14.1	0.22	0.14	0.21
Heritability		0.52	0.66	0.5	0.58

*, **, ***, significant at 0.05, 0.01, and 0.001 levels of probability, respectively: HSW, 100-seed weight; SL, seed length; SW, seed width; ST, seed thickness.

**Table 2 genes-13-02350-t002:** DArTseq SNPs markers with significant associations with seed size traits of 195 accessions of African yam bean evaluated during the 2018 and 2019 cropping seasons at three locations in Nigeria (Ibadan, Kano, and Ubiaja).

SN	Trait	Marker	Positions	GLM		MLM		Significant in Individual Locations
				*p*-Value	Marker R^2^	*p*-Value	Marker R^2^	
1	HSW	29421549|F|0-25:A>C-25:A>C	1124	1.14 × 10^−4^	0.08362	8.76 × 10^−4^	0.0754	KANO, UBIAJA
2	HSW	100026424|F|0-37:C>T-37:C>T	177	1.62 × 10^−4^	0.06499			UBIAJA
3	HSW	100008851|F|0-26:T>C-26:T>C	106	2.31 × 10^−4^	0.07655			
4	HSW	100041049|F|0-42:G>C-42:G>C	249	7.12 × 10^−4^	0.06635			
5	HSW	100026423|F|0-13:A>T-13:A>T	2228	7.12 × 10^−4^	0.06635			
6	HSW	29421658|F|0-5:G>A-5:G>A	1156	8.34 × 10^−4^	0.06495			
7	SL	29420365|F|0-55:C>G-55:C>G	2152	4.35 × 10^−4^	0.07298	8.84 × 10^−4^	0.0769	
8	ST	100034480|F|0-31:C>A-31:C>A	216	1.02 × 10^−5^	0.09925	3.42 × 10^−5^	0.1036	UBIAJA
9	ST	29420736|F|0-57:G>T-57:G>T	2364	4.59 × 10^−5^	0.08693	9.84 × 10^−5^	0.0925	IBADAN, KANO
10	ST	29420680|F|0-49:T>G-49:T>G	729	4.41 × 10^−4^	0.06803	7.61 × 10^−4^	0.0712	
11	ST	29420334|F|0-52:C>G-52:C>G	525	5.16 × 10^−4^	0.06702	7.09 × 10^−4^	0.0719	
12	SW	29420736|F|0-57:G>T-57:G>T	2364	1.40 × 10^−5^	0.10405	3.63 × 10^−5^	0.1084	KANO
13	SW	29421428|F|0-9:C>A-9:C>A	1088	1.19 × 10^−4^	0.08505	2.08 × 10^−4^	0.0891	KANO
14	SW	29420334|F|0-52:C>G-52:C>G	525	5.98 × 10^−4^	0.0708	7.94 × 10^−4^	0.0746	

HSW, hundred seeds weight; SL, seed length; SW, seed width; ST, seed thickness.

**Table 3 genes-13-02350-t003:** Significant markers whose nucleotide sequences were found on the *Phaseolus vulgaris* genome and the encoding protein of genes found close to them.

SN	Trait	Marker	Position	Gene ID	Crop	Chromosome	Encoding Product	Role	References
1	HSW	100026424|F|0-37:C>T-37:C>T	Pv06:15,126,920..15,126,980	Phvul.006G04320	*P. vulgaris*	6	Cinnamoyl-CoA reductase	Lignin biosynthesis in seed coat	[55]
				Phvul.006G043600	*P. vulgaris*	6	RING/U-box superfamily protein	Regulate seed size	[56,57,58]
				Phvul.006G043700	*P. vulgaris*	6	Calcium-dependent protein kinase 33	Seed development	[59,60]
				Phvul.006G044800	*P. vulgaris*	6	3-hydroxy-3-methylglutaryl-coenzyme A reductase-like protein	Seed development	[61]
				Phvul.006G045000	*P. vulgaris*	6	RING-H2 finger protein 2B	Seed development	[62]
				Phvul.006G045300	*P. vulgaris*	6	Myb transcription factor	Regulate seed size	[63,64]
				Phvul.006G047700	*P. vulgaris*	6	Pentatricopeptide repeat (PPR) superfamily protein	Regulate seed size	[65]
				Phvul.006G047300	*P. vulgaris*	6	WRKY family transcription factor	Regulate seed size	[65]
2	HSW	100041049|F|0-42:G>C-42:G>C	Pv06:15,580,580..15,580,630	Phvul.006G046700	*G. max*	6	GDP-L-galactose phosphorylase 1-like	Regulate seed size	[66]
				Phvul.006G047300	*P. vulgaris*	6	WRKY family transcription factor	Regulate seed size	[13,65]
				Phvul.006G047700	*P. vulgaris*	6	Pentatricopeptide repeat (PPR) superfamily protein	Regulate seed size	[65]
				Phvul.006G047900	*P. vulgaris*	6	zinc finger CCCH-type with G patch domain protein	Regulate seed size	[67]
				Phvul.006G050000	*P. vulgaris*	6	UDP-Glycosyltransferase superfamily protein	Seed development	[68]
				Phvul.006G051100	*P. vulgaris*	6	Transmembrane protein, putative	Regulate seed size	[63]
				Phvul.006G053300	*P. vulgaris*	6	WRKY family transcription factor family protein	Regulate seed size	[13,65]
				Phvul.006G053800	*P. vulgaris*	6	ATP-binding ABC transporter	Regulate seed size/weight	[69]
				Phvul.006G054100	*P. vulgaris*	6	Cytochrome P450 superfamily protein	Regulate seed/fruit size	[70]
3	ST	100034480|F|0-31:C>A-31:C>A	Pv02:2,303,744..2,303,815	Phvul.002G016300	*P. vulgaris*	2	Ubiquitin-conjugating enzyme 3	Regulate seed size	[15,56]
				Phvul.002G016400	*P. vulgaris*	2	UDP-glucosyltransferase family protein	Regulate grain size	[71]
				Phvul.002G016900	*G. max*	2	Ethylene-responsive transcription factor 3-like	Mediates seed size and seed weight	[72,73]
				Phvul.002G017600	*G. max*	2	Transcription factor SPATULA-like	Regulate grain size	[74]
				Phvul.002G019500	*P. vulgaris*	2	Cyclin-dependent kinase inhibitor family protein	Regulate seed size/weight	[75]
				Phvul.002G021600	*G. max*	2	Serine/threonine protein kinase TIO-like	Regulate seed size	[76]
				Phvul.002G024100	*P. vulgaris*	2	Pentatricopeptide repeat (PPR) superfamily protein	Regulate grain size	[65]
				Phvul.002G022600	*P. vulgaris*	2	GDSL-like Lipase/Acylhydrolase superfamily protein	Seed development	[77]
				Phvul.002G022800	*P. vulgaris*	2	Cytochrome P450 superfamily protein	Regulate seed/fruit size	[70]
				Phvul.002G023100	*P. vulgaris*	2	Transducin/WD40 repeat-like superfamily protein	Regulate grain size	[78]
4	SL	29420365|F|0-55:C>G-55:C>G	Pv01:40,806,541..40,806,608	Phvul.001G153000	*P. vulgaris*	1	Ovate family protein 13	Regulate seed/fruit size	[79,80]
				Phvul.001G152900	*P. vulgaris*	1	RING/FYVE/PHD zinc finger superfamily protein	Regulate seed size	[81]
				Phvul.001G153400	*P. vulgaris*	1	Kelch repeat F-box protein	Regulate seed size	[82]
				Phvul.001G153700	*P. vulgaris*	1	Pentatricopeptide repeat (PPR) superfamily protein	Regulate seed size	[65]
				Phvul.001G155600	*G. max*	1	β-carotene isomerase D27	Regulate seed length	[81]
				Phvul.001G156500	*P. vulgaris*	1	Auxin response factor 11	Regulate seed size	[12]
				Phvul.001G157400	*P. vulgaris*	1	E3 ubiquitin-protein ligase DRIP2-like	Regulate seed size	[56,83]
				Phvul.001G157600	*G. max*	1	Ethylene-responsive transcription factor 12-like	Regulate seed size	[72]
				Phvul.001G157900	*P. vulgaris*	1	Cytochrome P450 superfamily protein	Regulate seed/fruit size	[70]
5	ST, SW	29420736|F|0-57:G>T-57:G>T	Pv07:40,040,935..40,041,1	Phvul.007G269900	*P. vulgaris*	7	Ovate family protein 13	Regulate seed/fruit size	[79,80]
				Phvul.007G270100	*P. vulgaris*	7	Ubiquitin-conjugating enzyme 20	Regulate seed size	[15,56]
				Phvul.007G272700	*P. vulgaris*	7	RING-H2 finger protein 2B	Seed development	[62]
				Phvul.007G273100	*P. vulgaris*	7	Serine/threonine protein phosphatase 2A	Regulate grain shape	[76,84]
				Phvul.007G273400	*P. vulgaris*	7	Myb transcription factor	Regulate grain size	[63,64]
				Phvul.007G278500	*P. vulgaris*	7	Pentatricopeptide repeat (PPR) superfamily protein	Regulate seed size	[65]
				Phvul.007G278600	*P. vulgaris*	7	Argonaute family protein	Regulate seed size	[85]
				Phvul.007G279400	*P. vulgaris*	7	ARM repeat superfamily protein	Regulate seed size	[86]
				Phvul.007G280200	*P. vulgaris*	7	ATP-binding/protein serine/threonine kinase	Regulate seed size	[76]

HSW, hundred seeds weight; SL, seed length; SW, seed width; ST, seed thickness.

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
