# Peer review of "Genome-Wide Association Study Revealed SNP Alleles Associated with Seed Size Traits in African Yam Bean (Sphenostylis stenocarpa (Hochst ex. A. Rich.) Harms)"

_genes, 2022, doi:10.3390/genes13122350_

Round 1

Reviewer 1 Report

In this manuscript, the authors studied the Marker-trait association for seed-size traits in African yam bean [Sphenostylis stenocarpa (Hochst ex. A. Rich.) Harms]. This work was well done, and the methods of data collection were appropriate.

However, the authors should improve their paper according to the following comments:

The manuscript deals with "Marker-trait association for seed-size traits in African yam bean [Sphenostylis stenocarpa (Hochst ex. A. Rich.) Harms].

First: Title: It should change to the following:

1)   Genome-wide association mapping revealed SNP alleles associated with seed-size traits in African yam bean [Sphenostylis stenocarpa (Hochst ex. A. Rich.) Harms]

Second Abstract, keywords and Introduction:

2) has some minor corrections as in the attached file.

Third: The objectives of the study

3) is ok.

Fourth: Materials and Methods

4) has some minor corrections as in the attached file.

How many accessions were used in the current study 195 or 196? So, please check and correct the manuscript.

How about nitrogen and potassium fertilizers? (Dose and time of applications) also, minor elements?

Results and discussion

5)  has some minor corrections as in the attached file.

Figure 2. Phenotypic correlation coefficient of seed-size traits of 196 accessions of African 201 yam bean evaluated during the 2018 and 2019 cropping seasons at three locations in Ni-202 geria. 100-seed weight (HSW); Seed length (SL); Seed width (SW); Seed thickness (ST)

(How many accessions were used in the current study 195 or 196? So, please check and correct the manuscript)

Table 2. DArTseq SNPs markers have a significant association with seed-size traits of 195 260 accessions of African yam bean evaluated during the 2018 and 2019 cropping seasons at 261 three locations in Nigeria (Ibadan, Kano, and Ubiaja).

(How many accessions were used in the current study 195 or 196? So, please check and correct the manuscript)

References

6) the references are in a terrible status so, please check and correct some minor corrections as in the attached file.

Thank you for suggesting me as a reviewer for this paper.

with best regards

Author Response

Reviewer 1

Q1. Title: It should change to the following: ‘Genome-wide association mapping revealed SNP alleles associated with seed-size traits in African yam bean [Sphenostylis stenocarpa (Hochst ex. A. Rich.) Harms]’

Response:

Title has been changed as suggested.

Q2. Some minor corrections as in the attached file

 Response:

Corrections were made as suggested.

Q3. How many accessions were used in the current study 195 or 196? So, please check and correct the manuscript.

Response:

The experiment started with 196 accessions originally, and the same number was used for the phenotypic analysis. However, during SNP quality control, TSs-442 was filtered out due to low quality SNPs (lines 115-119), leaving us with 195 accessions for GWAS analysis.

Q4. How about nitrogen and potassium fertilizers? (Dose and time of applications) also, minor elements?

Response:

Phosphorus alone was applied based on the soil test result across environment (Olomitutu et al. 2022, https://doi.org/10.3390/agronomy12040884), the crop requirement and the protocol used by IITA. African yam bean is a legume that can fix nitrogen.  Phosphorus fertilizer application in the form of triple superphosphate at the rate of 50 kg P/ha, three weeks after planting (99-100).

Reviewer 2 Report

The authors need to explain the principal component analysis in detail. It is understood that they did not find any geographical correlation with the population structure/ PCA. However, they can make efforts to correlate the sub-groups with respect to their phenotype performance. Authors may also like to do further in-depth analysis for explaining the population structure.

Reviewer 3 Report

Line 12: markers

Line 12: Two pleiotropic markers: the effect of a marker is pleiotropic, not the marker itself. It should be “two markers with pleiotropic effects.” Please, correct all over the manuscript.

Line 15: what is seed density, and where is “100-seed weight”

Introduction line 82: please replace “In this work” which could refer to the work of Oluwole et al., with “Here”.

Line 102: 100-seed weight; this trait is not mentioned in the abstract. Please, add a measurement unit, e.g. mm or g, for all traits when possible.   

Line 141: the threshold is 10-4. However, figures 4 and 5 have a threshold of 10-3. Related to this point, figure 4 and table 2 show 14 SNPs above 10-3, and figure 5 shows 9 SNPs above 10-3. Please match the P values in the text and tables. How come the authors detected 58 SNP-trait associations, lines 205-206?

I strongly recommend reporting results from only MLM. The problem with GLM is that it doesn’t consider population structure. In line 183, the authors stated no population structure, but this was based on a limited number of SNPs, 2491.

Related to this point are lines 309-313; if the majority of accessions are possibly from Nigeria, I would expect a population structure. Again the inability to detect it can be due to the low number of SNPs. Therefore, GLM is not suitable.         

Line 148: why specifically common bean, any syntenic relationship? This is not explained.

Lines 238-244: I am confused; how can authors detect genes in less than 500 Kbp from the significant SNPs with an unknown genome?

Line 245: Why mention the chromosome number of the common bean? Syntenic relationships?

Lines 314-317: How the population structure presented here is different from the earlier paper that used a lower number of accessions (Reference 41)? 

Lines 326-327 are not valid. As the genome is not known, it could be that several SNPs correspond to the same gene.

In general, the discussion is weak and needs improvement. 

Round 2

Reviewer 2 Report

Authors need to check for minor English editing.

Author Response

Q1. Authors need to check for minor English editing

Respones.

The manuscript was read, and grammatical corrections were made as suggested.

Reviewer 3 Report

- 10-4 is used to refer to 0.0001, as 0.0009 is normally referred to as 0.001 10-3

- I wonder how the authors argue that "The less structured results of the GWAS population may be due to the GWAS population in which the majority of accessions either belong to Nigeria (90 accessions) or unknown (102 samples). These unknown samples show very close relatedness with Nigerian accession in both PCA and MDS population structure analysis approaches"

I wonder how 101 unknown accessions show close relatedness to the 90 Nigerian accessions, and the conclusion is no or less population structure?

- As a result of the population structure, MLM, not GLM, is the analysis to be used, otherwise, most of the detected SNPs might be false negative. 

Author Response

Q1. 10-4 is used to refer to 0.0001, as 0.0009 is normally referred to as 0.001 10-3.

Response:

Apologies for this; it should probably not be included in the comment. The threshold for P-values is 10-4. However, the Y-axis of the Manhattan plots states –log10(P). Meaning that the -log of P-values were calculated before the threshold lines were drawn.

For instance, if the P-value of a significant SNP for a trait is 8.76E-04. Then, the –log10(8.76E-04) which is equal to 3.0574 was used on the Manhattan plots. This was done for the smallest P-value for any significant SNP for a trait before the line was drawn.

Q2. I wonder how the authors argue that "The less structured results of the GWAS population may be due to the GWAS population in which the majority of accessions either belong to Nigeria (90 accessions) or unknown (102 samples). These unknown samples show very close relatedness with Nigerian accession in both PCA and MDS population structure analysis approaches"

I wonder how 101 unknown accessions show close relatedness to the 90 Nigerian accessions, and the conclusion is no or less population structure?

Response:

In the updated discussion following your previous suggestion (lines 329 and 330), the following statement was made as regards the subtle population structure that was found.

‘This might have resulted from the fact that most of the accessions, especially the large proportion whose origins are unknown (102 accessions), are possibly from Nigeria.’

Reasons:

In this study, IITA gene bank accessions were used. Various collectors collected these accessions. In some instances, the collectors failed to give the name of the exact location where the accessions were collected. As a result, the gene bank designated their origin as unknown. However, a review publication from the gene bank by Paliwal et al, (2020) suggested that ‘most of the 450 accessions of AYB landraces in the IITA gene bank were collected from different parts of Nigeria’. It is therefore possible that both Nigeria (90 samples) and the unknown (102 samples) share an ancestral relationship.

Reference

Paliwal, R.; Abberton, M.; Faloye, B.; Olaniyi, O. Developing the role of legumes in West Africa under climate change. Curr. Opin. Plant Biol. 2020, 56, 242–258.

Q3. As a result of the population structure, MLM, not GLM, is the analysis to be used, otherwise, most of the detected SNPs might be false negative.

Response:

In the discussion (lines 338–347) of the newly updated manuscript, the following statements were made.

‘Using two alternative GWAS models, we found that the MLM model had a stricter decrease in the number of significant markers than the GLM model. This is because GLM is considered a naive model with a high rate of false-positives because it does not population structure take account [91], whereas MLM takes population structure into account and avoids spurious associations [92,93]. Though subtle population structure was found in this study, both models were reported because this was the first attempt at dissecting the genetic basis of seed-size traits in AYB using GWAS. Significant QTLs associated with agronomic traits have also been reported in the absence of population structure using GLM and MLM models of GWAS in rice and faba bean crops [92,94].’

In this study, we had no clear population structure.